# Development and validation of the arabic version of the social-ecological model questionnaire for patients undergoing maintenance hemodialysis

**Ayman S. Abutair**[1], **Md Mizanur Rahman**[1]*, **Asri Bin Said**[2]

**1** Department of Community Medicine and Public Health, Faculty of Medicine and Health Sciences, Universiti Malaysia, Kota Samarahan, Sarawak, Malaysia, **2** Department of Medicine, Faculty of Medicine and Health Sciences, Universiti Malaysia, Kota Samarahan, Sarawak, Malaysia

* rmmizanur@unimas.my

## Abstract

### Background

Patients undergoing maintenance hemodialysis (MHD) in Arabic-speaking regions, particularly in Palestine, face unique sociopolitical and cultural barriers affecting their care and quality of life. Existing assessment tools rarely address the multi-level determinants of support within these populations, highlighting the need for a culturally validated instrument based on the Social-Ecological Model (SEM).

### Objective

To develop and validate an Arabic-language questionnaire grounded in the SEM to assess the multi-level (community, interpersonal, organizational, and policy) support factors influencing MHD patients in the Gaza Strip.

### Methods

A cross-sectional study was conducted from November 2024 to February 2025 at three governmental dialysis centers in Gaza, enrolling 101 Arabic-speaking adult MHD patients through systematic random sampling. The validation process encompassed item analysis, content and face validity, and psychometric testing, including exploratory factor analysis (EFA), assessment of convergent and discriminant validity which includes average variance extracted, factor loadings, composite reliability, Cronbach's alpha, Fornell-Larcker criterion, cross-loadings, and Heterotrait-Monotrait ratio.

### Results

The EFA identified a four-factor structure corresponding to Community, Interpersonal, Organizational, and Policy Support, explaining 65.15% of the variance. The final

**Data availability statement:** The data underlying the study's results are available from the following URL: https://figshare.com/s/5bdf493e766642acb43d (DOI: 10.6084/m9.figshare.29203493).

**Funding:** The author(s) received no specific funding for this work.

**Competing interests:** The authors have declared that no competing interests exist.

validated questionnaire comprised 31 items, with the Community Support domain demonstrating the strongest psychometric properties (factor loadings: 0.821–0.923; Cronbach's alpha = 0.972; AVE = 0.796). While Interpersonal and Organizational domains showed acceptable reliability and validity, the Policy Support domain displayed marginal construct validity. Overall, the instrument had strong content, face, convergent, and discriminant validity.

## Conclusion

The Arabic SEM-based questionnaire is a reliable and valid tool for assessing multi-level support factors among MHD patients in Arabic-speaking contexts. It enables comprehensive evaluation for research and clinical practice. Future research should confirm its structure using confirmatory factor analysis, extend validation to diverse Arab populations, and examine temporal stability.

## Introduction

End-stage renal disease (ESRD) represents the most advanced stage of chronic kidney disease, characterized by the permanent loss of renal function [1]. It is one of the leading causes of death and morbidity worldwide [2]. Maintenance hemodialysis (MHD) is a life-sustaining treatment for individuals with ESRD. It is commonly known that patients undergoing MHD experience a wide range of physical, emotional, and psychological challenges. These include fatigue, social isolation, treatment burden, and elevated psychological stress such as depression and anxiety, which may lead to a reduction in quality of life (QoL) and poor adherence to treatment regimens [3–6]. Social support networks, health literacy, and healthcare accessibility also play critical roles in determining patient outcomes among this group.

In this context, the Social-Ecological Model—first introduced by Bronfenbrenner in 1977 as an ecological systems theory and later adapted in public health to address multi-level health determinants—offers a robust conceptual framework for understanding the interplay of personal, social, institutional, and policy-level factors [7]. The SEM outlines five levels of influence: intrapersonal, interpersonal, organizational, community, and policy. It highlights how individuals interact with their environment and underscores the value of integrated, multi-level approaches to health promotion [8].

This model has demonstrated effectiveness in guiding public health interventions, including policy design and behavioral change in areas such as chronic disease management, cancer prevention, physical activity, dietary behaviors, mental health, and vaccination uptake [9–14]. In the context of MHD, SEM can help elucidate how environmental constraints, insufficient social support, and structural healthcare barriers affect treatment outcomes, mental well-being, and QoL. For instance, recent studies have employed SEM to investigate dialysis patients' decisions regarding kidney transplantation and factors influencing adherence to fluid restriction [15,16]. However, its application to MHD-specific contexts remains underdeveloped, particularly

in terms of assessment tools that systematically capture these multi-level influences. For example, Akinyemiju et al. [9] present a socio-ecological framework for cancer prevention in low- and middle-income countries, demonstrating that community and organisational support are crucial for cancer screening and ongoing care. Similarly, Lee et al. [15] demonstrate how policy-level interventions such as public awareness campaigns, alongside support from healthcare organizations, enhance fluid restriction adherence in hemodialysis populations. Furthermore, Tanhan & Francisco [13] find that multi-tiered interventions in psychosocial and mental health support systems lead to positive outcomes in chronic disease management, with implications directly relevant to MHD contexts.

Despite growing interest in patient-centered and culturally informed care, there is a lack of validated instruments tailored to Arabic-speaking MHD populations that comprehensively assess SEM-related determinants. The Arabic-speaking context, particularly in regions such as the Gaza Strip, is characterized by distinct sociopolitical, economic, and healthcare challenges—including chronic conflict, resource scarcity, and fragmented healthcare infrastructure—which intensify the burden of chronic illness [17]. These regional dynamics are seldom reflected in existing assessment tools, most of which are developed in Western contexts and lack cultural resonance.

Moreover, existing instruments such as the Medical Outcomes Study [18], Social Support Survey [19], and the Duke Social Support Index [20] focus narrowly on interpersonal or intrapersonal factors and do not adequately address SEM's broader contextual dimensions. These limitations highlight the urgent need for developing a culturally appropriate, SEM-based instrument that captures the diverse influences affecting MHD patients in Arabic-speaking communities. By integrating various layers of influence, including individual, interpersonal, community, and societal factors, the SEM helps capture a broad and nuanced understanding of the context. This holistic approach not only considers intrapersonal dynamics but also the broader environmental and societal conditions that may impact outcomes. Such an inclusive perspective enables a more thorough assessment and better-informed decision-making.

To address this gap, this study aims to develop and validate an Arabic version of an SEM-based questionnaire specifically designed for patients undergoing MHD. This tool would facilitate a better understanding of the contextual challenges patients face and support the design of more effective, culturally tailored interventions. In this way, it would enhance research and clinical care in Arabic-speaking healthcare systems, ultimately leading to improved quality of life and patient-centred care for individuals with ESRD.

## Materials and Methods

### Design

This study employed a cross-sectional design to develop and validate an Arabic version of the SEM questionnaire tailored for patients undergoing MHD. The research was conducted across all governmental hemodialysis centers in the Gaza Strip, Palestine, including Al-Shifa Medical Complex, Nasser Medical Complex, and Al-Aqsa Hospital. These centers collectively provide services to the majority of patients with ESRD in the Gaza Strip, making them ideal settings for capturing a representative sample of the MHD population. The study was conducted between November 3, 2024, and February 13, 2025. Data collection took place over several weeks, utilising a systematic sampling method to recruit participants attending their routine dialysis sessions during the study period. The validation of a questionnaire is a multifaceted process that ensures the instrument is reliable and valid for its intended purpose [21]. This process can be divided into several key stages: development, translation, content validation, and psychometric analysis (Fig 1).

### Participants and sampling method

A total of 101 adult patients (≥18 years) with ESRD undergoing MHD were recruited from three governmental dialysis centers across the Gaza Strip. Patients who were undergoing MHD and were at least 19 years old, had been undergoing MHD for at least three months, and spoke and wrote in Arabic were included in the study. While, patients under peritoneal dialysis or

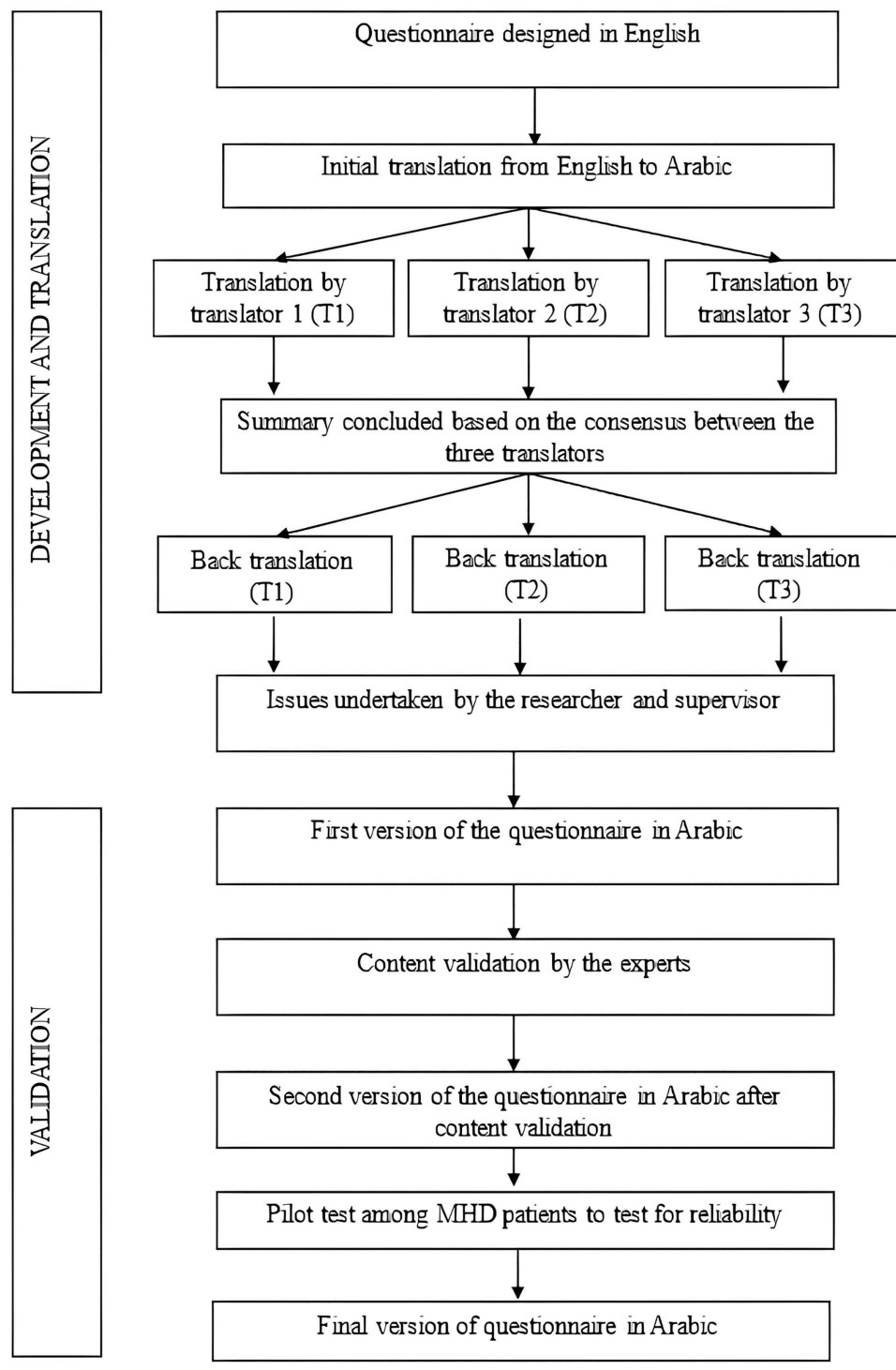

**Fig 1. Summary of the development of the questionnaire.**

kidney transplant, hospitalization at the time of recruitment, were under intensive care or palliative care treatment, patients under dietitian supervision and/or nutritional support (such as patients on Ryle's tube feeding or ONS), any patients diagnosed with a major illness (significant liver diseases, gastrointestinal disorders, cancer requiring chemotherapy, severe CHF, chronic pulmonary diseases, or other considerable disorders), patients not fit to complete the study protocols or had communication barriers or difficulties even with the help of a caregiver, and patients who refuse to sign consent were excluded from the study.

A systematic random sampling method was employed to select participants. Participants were selected during their routine dialysis sessions. The study aims to sample roughly 8–10 patients daily. The data collection process was conducted from four to five days per week. On each data collection day, 65 patients from Al-Shifa Medical Complex, 55 patients from Nasser Medical Complex, and 15 patients from Al-Aqsa Hospital (or the closest integer) who attended these dialysis centres for HD sessions were selected. The sample size for this study was determined based on a statistical saturation approach for psychometric scale validation studies, rather than a priori power analysis or participants-per-item formulas. Data collection and analysis were conducted iteratively, with periodic assessment of internal consistency reliability (Cronbach's alpha) and factor structure stability. Once statistical indicators, such as Cronbach's alpha and factor loadings, reached acceptable thresholds and further data collection did not appreciably improve these metrics, recruitment was stopped. This pragmatic approach is recognised in the instrument development literature, as it balances resource considerations with methodological rigour, ensuring the final sample size supports stable and interpretable psychometric properties. The resulting sample size met the recommended reliability and validity standards for exploratory factor analysis and scale validation in health research contexts, reaching the desired daily sample size. The starting patient was selected randomly using a simple random method.

## Conceptual framework

The SEM includes five levels; however, in designing this questionnaire, we intentionally excluded the intrapersonal level and focused on external levels of influence (interpersonal, organizational, community, and policy) based on both theoretical and practical considerations. Two main factors guided the exclusion. First, the respondent burden. The patients undergoing MHD often experience fatigue and cognitive strain due to prolonged treatment sessions [4,6]. Including the intrapersonal level would have significantly lengthened the questionnaire, potentially reducing completion rates and data quality. Second, we excluded the intrapersonal level to prioritise external factors (interpersonal, organisational, community, and policy) that critically influence the experiences of MHD patients, especially in Arabic-speaking populations, where family and community support play a significant role. Previous research suggests that external environmental and social support systems play a more critical role in shaping treatment adherence, psychological well-being, and QoL in chronic conditions than individual traits alone [16,22]. Future studies could explore intrapersonal factors qualitatively or through socio-demographic variables (e.g., age, gender, education) as proxies for individual-level influences [23].

## Questionnaire development

The research team assessed the participants' responses to SEM levels in developing the questionnaire. The questionnaire was designed to cover four levels of SEM, including assessment at the interpersonal, organizational, community, and policy levels. The interpersonal, organizational, community, and policy levels were addressed with a specific number of items (10 items for each). The initial questionnaire was developed in English drawing on diverse sources for comprehensive content coverage. Three native Arabic speakers independently conducted forward translation; all translators were either healthcare or language professionals with expertise in clinical and public health contexts or linguistics. The research team reconciled variations among the translations through discussion, and the synthesized Arabic version was then back-translated into English by a separate language expert to check for equivalence. Any discrepancies identified during this process were jointly reviewed and resolved by the principal investigator, supervisor, and an additional linguistic expert to ensure semantic and cultural accuracy. Panel review and expert input from both clinical and language specialists further supported adaptation and content validation.

## Content validation

Content validation is a crucial procedure that ensures the relevance, clarity, simplicity, and absence of ambiguity in targeted items. Within this research, the content validation process engaged the expertise of nine proficient assessors, encompassing three linguistic specialists and six researchers (health professionals and health educators), of whom two were public health researchers, two were clinical nutritionists with experience in dialysis care, one was a nephrologist, and one was a health education specialist. These experts have extensive knowledge pertinent to the subject matter under investigation. Each item was evaluated based on four fundamental attributes: relevance to the specific domains, content clarity, presentation simplicity, and degree of ambiguity. The assessment was structured using a four-point scale, wherein each attribute was appraised and scored, with a rating of (1) indicating the item as 'Not relevant/Not clear/Not simple/ Doubtful,' and (4) signifying that the item was deemed 'Very relevant/Very clear/Very simple/Meaning is clear.'

Item-level content validity refers to the extent to which individual items on a measurement scale accurately represent the intended construct they are designed to measure. This is typically quantified using the Item-Level Content Validity Index (I-CVI), which is calculated by dividing the number of experts who rate an item as relevant (usually on a scale of 1–4) by the total number of experts. A commonly accepted cut-off value for I-CVI is 0.70, indicating that at least 70% of experts agree on the relevance of an item. At the same time, a higher threshold, such as 0.78, is sometimes recommended for greater rigour in specific contexts [24–26]. In addition to I-CVI, the Scale-Level Content Validity Index (S-CVI) was employed to assess the overall validity of the scale. The S-CVI was calculated using the average method (S-CVI/Ave) and the universal agreement method (S-CVI/UA). A typical cut-off value for the S-CVI is 0.80, suggesting that the overall scale is considered valid when at least 80% of experts agree on the relevance of items [27–29]. The study assessed content validity using the I-CVI, S-CVI/Ave, and S-CVI/UA to evaluate the relevance, clarity, simplicity, and ambiguity of the content. For six experts, a CVI of 0.83 was deemed acceptable, aligning with the established content validity thresholds [30]. These indices provide a systematic approach to evaluating the content validity of measurement tools, ensuring that relevant and representative of the constructs.

## Face validation

An instrument's effectiveness and appropriateness for its intended use depend on its face validity, which is evaluated for a variety of aspects, including grammar and language, clarity, relevance, cultural appropriateness, readability, question layout, item redundancy, comprehensiveness, response options, and time to complete [31]. Thirty patients participated in the evaluation process and used a five-point Likert scale (ranging from "strongly disagree" to "agree strongly") to provide comments on ten distinct categories. In survey research, this method of evaluation is well known for gathering subjective thoughts and opinions about the layout and content of the instrument [32]. The intraclass correlation coefficient (ICC) for face validity in this study was computed using a two-way mixed-effects model (ICC[3,k]), assessing absolute agreement among raters for each item based on Likert-scale ratings. Both single- and average-measure ICCs are reported, with the single-measure ICC at 0.720 (95% CI: 0.605–0.828) and the average-measure ICC at 0.963 (95% CI: 0.939–0.980), reflecting the reliability of individual raters and the overall panel, respectively [33]. According to established benchmarks, an ICC between 0.75 and 0.90 indicates "good" agreement, and values above 0.90 are considered "excellent," supporting the robustness and consistency of the scale's face validity evaluation in this context [31].

## Data preparation

All collected questionnaires were manually reviewed for completeness and accuracy. Any duplicate entries or inconsistent responses were identified and resolved before data entry. The data were transcribed into a Microsoft Excel spreadsheet, incorporating validation mechanisms to minimize entry errors. The raw data were then imported into the Statistical Package for Social Sciences, version 29 for Windows, for comprehensive statistical analysis. Data cleaning procedures

involved reviewing each questionnaire for completeness at the time of collection and again before analysis. Item analysis was conducted to examine the performance of individual items. No questionnaire with missing responses on SEM items was observed from the psychometric analysis, including EFA and validity testing. The final dataset was screened for duplication, inconsistencies, and missing values. No missing data were present in the variables included in the EFA and validation steps. These steps are essential for maintaining the integrity and reliability of the research findings.

**Psychometric analysis**

Psychometric analysis was assessed through a multi-step psychometric evaluation process, including exploratory factor analysis (EFA), convergent validity, and discriminant validity.

1. Exploratory Factor Analysis

To evaluate the psychometric analysis of the Arabic version of the SEM questionnaire, EFA was performed. EFA was chosen to explore the underlying factor structure and to assess the alignment of questionnaire items with the intended support domains. Before conducting EFA, assumptions of normality, sampling adequacy, and inter-item correlation were tested to confirm suitability for factor analysis.

The factorability of the data was assessed using the Kaiser-Meyer-Olkin (KMO) measure of sampling adequacy and Bartlett's Test of Sphericity. A KMO value above 0.60 and a significant Bartlett's test ($p < .05$) were considered necessary to proceed. A principal axis factoring method with oblique rotation (Oblimin) was applied, as the factors were expected to be correlated based on the theoretical assumptions of the SEM framework.

The number of factors to retain was determined using a combination of criteria, including eigenvalues greater than one, scree plot inspection, parallel analysis, and theoretical interpretability. Items were evaluated based on factor loading values, conceptual clarity, and the absence of significant cross-loadings. Items with low loadings (<.40), redundancy, or poor conceptual alignment were considered for removal to refine the scale structure. All relevant assumptions and criteria were applied to ensure methodological rigor. The final structure was intended to align with four external domains of the SEM.

2. Convergent Validity

Convergent validity was evaluated by examining factor loadings, Average Variance Extracted (AVE), Cronbach's alpha, and composite reliability for each domain. A factor loading ≥ 0.60, AVE ≥ 0.50, Cronbach's alpha ≥ 0.70, and composite reliability ≥ 0.70 were considered acceptable indicators of convergent validity. Items not meeting the threshold for loading were reviewed for potential removal. AVE values were used to assess the proportion of variance captured by the construct in relation to measurement error.

**Discriminant validity.** Discriminant validity was assessed using three complementary methods; 1) Fornell-Larcker criterion, whereby the square root of AVE for each construct must exceed its correlation with other constructs, 2) Cross-loadings, to ensure that each item loads more strongly on its associated construct than on others, 3) Heterotrait-Monotrait Ratio (HTMT), with values < 0.90 indicating acceptable discriminant validity.

**Ethical issues**

Given the potential ethical concerns associated with this study, it is crucial to emphasize that respondent participation was entirely voluntary. Throughout the research process, stringent measures were taken to ensure the confidentiality and privacy of the respondents, including the protection of their personal information. Importantly, ethical approval was obtained from the Medical Research Ethics Committee of Universiti Malaysia Sarawak (Ref: FME/24/136) and the local Helsinki Committee (PHRC/HC/1369/24), highlighting the commitment to maintaining high ethical standards in the conduct of the research. Informed consent was obtained from all participants before their involvement in the study. Written informed consent was provided and signed by each participant.

 

## Results

### Characteristics of respondents

This study included 101 adult patients undergoing MHD, recruited from three governmental dialysis centers in the Gaza Strip. The respondents were adult patients under MHD from the Gaza Strip, Palestine, aged 20–78, with a majority (n = 57) of females (56.4%). Most respondents (n = 74) were married (73.3%) and (78.2%) had a low education level (n = 79). This demographic profile reflects a diverse sample of Gazan patients under MHD.

### Preliminary item analysis

The developed questionnaire had four domains. Based on the feedback from experts and respondents, no items were removed from the questionnaire, and no items needed to be rephrased. The questionnaire includes 10 items for each of the four SEM domains, ensuring balanced representation. Interpersonal items focus on emotional, practical, and informational support. Organizational items address healthcare systems' roles, including education, empathy, family involvement, and mental health services. Community items capture emotional, practical, educational, and social support. Policy items examine access to treatment, medical supplies, trained personnel, psychological and nutritional support, awareness, patient rights, coordination, equity, and responsiveness to patient needs—all critical for patients undergoing MHD. However, during the EFA, 4 items were removed due to redundancy or low factor loadings: one item from the Interpersonal domain, two from the Organizational domain, and one from the Policy domain. In the subsequent assessment of convergent validity, an additional 5 items were removed due to factor loadings below 0.60: three from the Interpersonal domain, one from Organizational, and one from Policy Support. After these stages, the final validated questionnaire consisted of 31 items: 6 in Interpersonal Support, 7 in Organizational Support, 8 in Policy Support, and 10 in Community Support.

### Content validity

The content analysis of the questionnaire evaluates four parameters: relevance, clarity, simplicity, and ambiguity, using three indices: I-CVI, S-CVI, and S-CVI/UA. For the parameters of relevance, clarity, simplicity, and ambiguity, the I-CVI ranges from 0.83 to 1.00, indicating that experts generally deem individual items highly relevant and clear, consider individual items simple, and mostly view items as unambiguous. The S-CVI ranged from 0.98 to 1.00, suggesting that, on average, the scale was considered almost entirely relevant, the scale is perceived as very clear overall, the scale is simple on average, and the scale is perceived as generally unambiguous. The S-CVI/UA also ranged from 0.90 to 1.00, reflecting a high level of universal agreement among experts on the relevance of the scale items, demonstrating substantial universal agreement on item clarity, indicating a high universal agreement on the simplicity of the items, and indicating a significant level of universal agreement on the lack of ambiguity in the items. The questionnaire items appear well-constructed, with strong content validity in relevance, clarity, simplicity, and ambiguity, as indicated by the high values obtained across all indices and parameters (Table 1).

### Face validity

Table 2 reports face validity results using a two-way mixed-effects ICC model for the questionnaire. A single-measure ICC of 0.720 (95% CI: 0.605–0.828) indicates moderate agreement among individual raters, while an average-measure ICC of 0.963 (95% CI: 0.939–0.980) demonstrates excellent reliability when considering the aggregate ratings of all raters. Both results are statistically significant (p < 0.001), confirming that the questionnaire demonstrates strong reliability and consistency in terms of item clarity and content as assessed by the panel.

### Exploratory factor analysis

The factor analysis identified four distinct components representing critical support domains for MHD patients, which explained 67.49% of the variance. Through the use of Oblimin Rotation, a clear and simple structure was achieved, where

**Table 1. Content analysis of the questionnaire.**

| Parameters | I-CVI | S-CVI | S-CVI/UA |
|---|---|---|---|
| Relevance | 0.83-1.00 | 0.98-1.00 | 0.90-1.00 |
| Clarity | 0.83-1.00 | 0.98-1.00 | 0.90-1.00 |
| Simplicity | 0.83-1.00 | 0.98-1.00 | 0.90-1.00 |
| Ambiguity | 0.83-1.00 | 0.98-1.00 | 0.90-1.00 |

I-CVI = item-level content validity

S-CVI = scale-level content validity based (on average)

S-CVI/UA = scale-level content validity based on universal agreement

**Table 2. Face validity of the questionnaire.**

| Measures | Intraclass Correlation | 95% CI | | F Test with True Value 0 | | | |
|---|---|---|---|---|---|---|---|
| | | LL | UL | Value | df1 | df2 | p-value |
| Single | 0.720 | 0.605 | 0.828 | 28.637 | 29 | 261 | <0.001 |
| Average | 0.963 | 0.939 | 0.980 | 28.637 | 29 | 261 | <0.001 |

LL= Lower limit of 95% confidence interval

UL = Upper limit of 95% confidence interval

*p < .05, **p < .01, ***p < .001

each component corresponded to a specific support type (Table 3). For item selection, item 8 from Family and Interpersonal Support was removed due to redundancy; in Organizational Support, items 5 and 8 were removed; and in Policy Support, item 10 was removed due to low loadings. The suitability of the sample for factor analysis was suggested by the Kaiser-Meyer-Olkin measure of 0.811, which indicated a considerable degree of shared variance among the variables. Bartlett's Test of Sphericity was highly significant (p < .001), signifying that the correlation matrix was not an identity matrix, and there were enough inter-correlations among the variables to justify factor analysis. The Measures of Sampling Adequacy (MSA) for individual items ranged from 0.596 to 0.924. Most variables displayed meritorious to excellent sampling adequacy (MSA > 0.80), demonstrating strong shared variance and suitability for factor analysis. Overall, the commonalities indicated the extracted components provided a robust representation for most variables, with values ranging from 0.530 to 0.971, indicating that each item's variance explained by the four factors varies from moderate (53%) to very high (97%) across items.

**Community domain.** The community domain shows strong correlations with the first principal component, indicating that community-related factors are highly associated with this component. For example, Com_1, Com_2, Com_3, Com_4, Com_5, Com_6, Com_7, Com_8, Com_9, and Com_10 have high factor loadings ranging from 0.786 to 0.952 on the first component. This suggests that community-related variables are closely linked and collectively contribute significantly to the underlying structure of the data. The other components have relatively lower loadings for these community variables, indicating their primary influence on the first component.

**Interpersonal domain.** The interpersonal domain exhibits a different pattern. The factor loadings for Int_1 to Int_10 on the second principal component are relatively high, ranging from 0.693 to 0.981. This indicates that interpersonal factors are strongly associated with the second component. The loadings on the other components are generally lower, suggesting that the interpersonal domain has a distinct and significant influence on the second component, separate from the other domains.

**Policy support domain.** The policy support domain shows strong correlations with the third principal component. Variables such as Pol_1, Pol_2, Pol_3, Pol_4, Pol_5, Pol_6, Pol_7, Pol_8, and Pol_9 have high factor loadings ranging

**Table 3. Exploratory Factor analysis (N = 101).**

| Items | Interpersonal | Community | Organizational Support | Policy support | Communalities | MSA |
|---|---|---|---|---|---|---|
| Int_1 | **0.979** | 0.007 | 0.044 | 0.051 | .971 | .809 |
| Int_2 | **0.959** | −0.026 | 0.045 | 0.042 | .911 | .877 |
| Int_3 | **0.702** | 0.166 | −0.102 | −0.146 | .892 | .786 |
| Int_4 | **0.693** | 0.033 | −0.111 | −0.006 | .705 | .857 |
| Int_5 | **0.714** | 0.149 | −0.085 | −0.142 | .885 | .797 |
| Int_6 | **0.972** | −0.041 | 0.065 | 0.074 | .927 | .843 |
| Int_7 | **0.981** | −0.029 | 0.085 | 0.033 | .939 | .857 |
| Int_9 | **0.980** | −0.013 | 0.009 | 0.035 | .961 | .915 |
| Int_10 | **0.953** | −0.025 | 0.022 | 0.070 | .913 | .920 |
| Com_1 | −0.075 | **0.935** | −0.014 | 0.032 | .855 | .865 |
| Com_2 | −0.024 | **0.897** | 0.045 | −0.033 | .848 | .730 |
| Com_3 | −0.027 | **0.952** | −0.020 | 0.046 | .890 | .760 |
| Com_4 | 0.056 | **0.854** | −0.101 | −0.017 | .754 | .845 |
| Com_5 | 0.041 | **0.885** | 0.069 | 0.042 | .867 | .888 |
| Com_6 | −0.010 | **0.857** | 0.007 | −0.033 | .782 | .924 |
| Com_7 | 0.031 | **0.870** | 0.044 | 0.086 | .865 | .869 |
| Com_8 | 0.006 | **0.827** | 0.115 | −0.045 | .835 | .910 |
| Com_9 | 0.129 | **0.786** | 0.140 | −0.042 | .886 | .776 |
| Com_10 | 0.112 | **0.845** | 0.025 | −0.022 | .869 | .803 |
| Org_1 | 0.082 | 0.023 | **0.602** | 0.003 | .563 | .703 |
| Org_2 | −0.075 | −0.037 | **0.811** | −0.045 | .855 | .692 |
| Org_3 | −0.060 | −0.082 | **0.830** | 0.177 | .798 | .781 |
| Org_4 | −0.054 | 0.124 | **0.694** | 0.041 | .698 | .756 |
| Org_6 | 0.110 | 0.010 | **0.547** | −0.113 | .530 | .715 |
| Org_7 | 0.072 | 0.082 | **0.670** | −0.062 | .723 | .806 |
| Org_9 | −0.009 | 0.260 | **0.587** | −0.078 | .662 | .794 |
| Org_10 | −0.110 | −0.021 | **0.763** | −0.080 | .845 | .713 |
| Pol_1 | 0.079 | −0.010 | 0.007 | **0.657** | .650 | .596 |
| Pol_2 | −0.085 | 0.174 | 0.038 | **0.724** | .728 | .736 |
| Pol_3 | 0.130 | −0.078 | −0.032 | **0.642** | .562 | .745 |
| Pol_4 | −0.020 | −0.057 | −0.062 | **0.749** | .687 | .736 |
| Pol_5 | −0.153 | −0.081 | 0.059 | **0.621** | .627 | .793 |
| Pol_6 | −0.034 | 0.003 | 0.035 | **0.779** | .759 | .684 |
| Pol_7 | −0.035 | 0.202 | −0.114 | **0.655** | .570 | .721 |
| Pol_8 | 0.097 | −0.112 | −0.052 | **0.704** | .744 | .715 |
| Pol_9 | 0.080 | −0.022 | −0.007 | **0.775** | .706 | .736 |

Extraction Method: Principal Component Analysis.

Rotation Method: Oblimin with Kaiser Normalization.

a. Rotation converged in 5 iterations.

from 0.621 to 0.779 on the third component. This suggests that policy support-related factors are highly associated with this component. The other components have relatively lower loadings for these policy support variables, indicating their primary influence on the third component.

**Organizational support domain.** The organizational support domain is strongly associated with the fourth principal component. Variables Org_1, Org_2, Org_3, Org_4, Org_6, Org_7, Org_9, and Org_10 have high factor loadings ranging from 0.547 to 0.830 on the fourth component. This indicates that organizational support-related factors are highly associated with this component. The other components have relatively lower loadings for these organizational support variables, suggesting that their primary influence is on the fourth component.

## Convergent validity

Convergent validity is a crucial aspect of construct validation, ensuring that items designed to measure a particular construct are indeed related to it. This analysis assessed the convergent validity of support domains for MHD patients, including Community, Interpersonal, Organizational, and Policy support (Table 4). However, certain items were removed due to low loadings: from the Interpersonal domain, items 2, 4, and 10; from the Organizational domain, item 1; and from the Policy domain, item 2. The remaining items were evaluated for their factor loadings, internal consistency reliability (Cronbach's alpha), composite reliability (rho_a and rho_c), and AVE to determine the strength of each domain's construct validity.

**Community support domain.** The Community Support domain demonstrates strong convergent validity, with factor loadings ranging from 0.821 to 0.923, indicating that the items are highly correlated with the underlying construct. The Cronbach's alpha value of 0.972 and the composite reliability (rho_a) of 0.995 both indicate excellent internal consistency reliability. The composite reliability (rho_c) is also very high at 0.975, further reinforcing the construct's reliability. The AVE is 0.796, which is well above the commonly accepted threshold of 0.5, suggesting that the construct explains a significant amount of variance in the items. Overall, the Community Support domain shows robust convergent validity and reliability.

**Interpersonal support domain.** The Interpersonal Support domain exhibits strong convergent validity, with factor loadings ranging from 0.613 to 0.996. Items such as Int_3 and Int_5 have particularly high loadings, indicating strong indicators of the construct. The Cronbach's alpha value of 0.962 and the composite reliability (rho_a) of 0.881 both suggest excellent internal consistency reliability. The composite reliability (rho_c) is also high at 0.901, further supporting the construct's reliability. The AVE is 0.614, which is above the threshold of 0.5, indicating that the construct explains a significant amount of variance in the items. Overall, the Interpersonal Support domain shows strong convergent validity and reliability.

**Organizational support domain.** The Organizational Support domain shows moderate to strong convergent validity, with factor loadings ranging from 0.619 to 0.844. The Cronbach's alpha value of 0.857 and the composite reliability (rho_a) of 0.854 both indicate good internal consistency reliability. The composite reliability (rho_c) is also high at 0.884, reinforcing the construct's reliability. However, the AVE is 0.526, which is slightly below the threshold of 0.5, suggesting that the construct explains a moderate amount of variance in the items. This indicates that while the Organizational Support domain is generally reliable, it may benefit from further refinement or additional items to improve its explanatory power.

**Policy support domain.** The Policy Support domain exhibits moderate to strong convergent validity, with factor loadings ranging from 0.562 to 0.833. The Cronbach's alpha value of 0.860 and the composite reliability (rho_a) of 0.895 both suggest good internal consistency reliability. The composite reliability (rho_c) is also high at 0.887, further supporting the construct's reliability. However, the AVE is 0.501, which is just satisfy the threshold of 0.5, indicating that the construct explains a moderate amount of variance in the items, although two items loadings were less than.60. This suggests that the Policy Support domain may need further refinement or additional items to improve its explanatory power.

## Discriminant validity

The discriminant validity of the four constructs (Community, Interpersonal, Organizational, and Policy Support) was assessed using the Fornell-Larcker criterion, cross-loadings, and HTMT ratios (Table 5). Results indicate that all constructs are distinct and meet recommended thresholds for discriminant validity, though minor refinements may enhance clarity.

**Table 4. Convergent validity.**

| Items | Loadings | Cronbach's alpha | Composite reliability (rho_a) | Composite reliability (rho_c) | Average variance extracted (AVE) |
|---|---|---|---|---|---|
| Com_1 | 0.872 | 0.972 | 0.995 | 0.975 | 0.796 |
| Com_2 | 0.911 | | | | |
| Com_3 | 0.914 | | | | |
| Com_4 | 0.821 | | | | |
| Com_5 | 0.923 | | | | |
| Com_6 | 0.856 | | | | |
| Com_7 | 0.888 | | | | |
| Com_8 | 0.899 | | | | |
| Com_9 | 0.917 | | | | |
| Com_10 | 0.913 | | | | |
| Int_1 | 0.647 | 0.962 | 0.881 | 0.901 | 0.614 |
| Int_3 | 0.995 | | | | |
| Int_5 | 0.996 | | | | |
| Int_6 | 0.613 | | | | |
| Int_7 | 0.696 | | | | |
| Int_9 | 0.649 | | | | |
| Org_2 | 0.844 | 0.857 | 0.854 | 0.884 | 0.526 |
| Org_3 | 0.619 | | | | |
| Org_4 | 0.629 | | | | |
| Org_6 | 0.676 | | | | |
| Org_7 | 0.759 | | | | |
| Org_9 | 0.692 | | | | |
| Org_10 | 0.822 | | | | |
| Pol_1 | 0.576 | 0.860 | 0.895 | 0.887 | 0.501 |
| Pol_3 | 0.695 | | | | |
| Pol_4 | 0.833 | | | | |
| Pol_5 | 0.661 | | | | |
| Pol_6 | 0.682 | | | | |
| Pol_7 | 0.562 | | | | |
| Pol_8 | 0.793 | | | | |
| Pol_9 | 0.807 | | | | |

**Community support domain.** The Community Support domain demonstrates strong discriminant validity. The Fornell-Larcker criterion shows that the square root of the AVE for Community Support (0.892) is higher than the correlations with other domains (Interpersonal: 0.356, Organizational: 0.454, policy: −0.181). This indicates that the Community Support construct is distinct from the other domains. Additionally, the cross-loadings for Community Support items (Com_1 to Com_10) are highest in their domain, further supporting discriminant validity. The HTMT ratio for Community Support with other domains is also below the threshold of 0.9, indicating that the constructs are empirically distinct.

**Interpersonal support domain.** The Interpersonal Support domain also shows good discriminant validity. The Fornell-Larcker criterion indicates that the square root of the AVE for Interpersonal Support (0.783) is higher than its correlations with other domains (Community: 0.356, Organizational: 0.019, policy: −0.171). The cross-loadings for Interpersonal Support items (Int_1, Int_3, Int_5, Int_6, Int_7, Int_9) are highest on their domain, reinforcing the discriminant validity.

**Table 5. Discriminant validity.**

| Items/Criteria | Community | Interpersonal | Organizational | Policy |
|---|---|---|---|---|
| **Fornell-Larcker criterion** | | | | |
| Community | **0.892** | | | |
| Interpersonal | 0.356 | **0.783** | | |
| Organizational | 0.454 | 0.019 | **0.725** | |
| Policy | −0.181 | −0.171 | −0.316 | **0.708** |
| **Cross-loadings** | | | | |
| Com_1 | **0.872** | 0.367 | 0.399 | −0.117 |
| Com_2 | **0.911** | 0.357 | 0.424 | −0.184 |
| Com_3 | **0.914** | 0.384 | 0.385 | −0.109 |
| Com_4 | **0.821** | 0.440 | 0.300 | −0.129 |
| Com_5 | **0.923** | 0.289 | 0.428 | −0.120 |
| Com_6 | **0.856** | 0.332 | 0.374 | −0.148 |
| Com_7 | **0.888** | 0.239 | 0.378 | −0.081 |
| Com_8 | **0.899** | 0.231 | 0.471 | −0.215 |
| Com_9 | **0.917** | 0.286 | 0.448 | −0.213 |
| Com_10 | **0.913** | 0.301 | 0.375 | −0.170 |
| Int_1 | 0.374 | **0.647** | −0.032 | −0.003 |
| Int_3 | 0.362 | **0.995** | 0.017 | −0.164 |
| Int_5 | 0.356 | **0.996** | 0.013 | −0.162 |
| Int_6 | 0.332 | **0.613** | −0.036 | 0.031 |
| Int_7 | 0.362 | **0.696** | −0.003 | −0.012 |
| Int_9 | 0.343 | **0.649** | −0.066 | −0.000 |
| Org_2 | 0.281 | −0.096 | **0.844** | −0.249 |
| Org_3 | 0.227 | −0.167 | **0.619** | −0.022 |
| Org_4 | 0.389 | −0.063 | **0.629** | −0.139 |
| Org_6 | 0.289 | 0.040 | **0.676** | −0.273 |
| Org_7 | 0.368 | 0.195 | **0.759** | −0.220 |
| Org_9 | 0.492 | 0.089 | **0.692** | −0.231 |
| Org_10 | 0.264 | −0.071 | **0.822** | −0.265 |
| Pol_1 | −0.073 | 0.049 | −0.142 | **0.576** |
| Pol_3 | −0.113 | 0.006 | −0.220 | **0.695** |
| Pol_4 | −0.178 | −0.148 | −0.305 | **0.833** |
| Pol_5 | −0.183 | −0.313 | −0.145 | **0.661** |
| Pol_6 | −0.098 | −0.143 | −0.155 | **0.682** |
| Pol_7 | 0.034 | 0.038 | −0.156 | **0.562** |
| Pol_8 | −0.168 | −0.136 | −0.325 | **0.793** |
| Pol_9 | −0.082 | −0.077 | −0.242 | **0.807** |
| **HTMT-Ratio** | | | | |
| Community | – | | | |
| Interpersonal | 0.401 | – | | |
| Organizational | 0.482 | 0.151 | – | |
| Policy | 0.172 | 0.141 | 0.309 | – |

The HTMT ratio for Interpersonal Support with other domains is below the threshold of 0.9, further confirming that the constructs are distinct.

**Organizational support domain.** The Organizational Support domain exhibits adequate discriminant validity. The Fornell-Larcker criterion shows that the square root of the AVE for Organizational Support (0.725) is higher than its correlations with other domains (Community: 0.454, Interpersonal: 0.019, policy: −0.316). The cross-loadings for Organizational Support items (Org_2, Org_3, Org_4, Org_6, Org_7, Org_9, Org_10) are highest on their domain, supporting discriminant validity. The HTMT ratio for Organizational Support with other domains is below the threshold of 0.9, indicating that the constructs are empirically distinct.

**Policy support domain.** The Policy Support domain demonstrates strong discriminant validity. The Fornell-Larcker criterion indicates that the square root of the AVE for Policy Support (0.708) is higher than its correlations with other domains (Community: −0.181, Interpersonal: −0.171, Organizational: −0.316). The cross-loadings for Policy Support items (Pol_1, Pol_3, Pol_4, Pol_5, Pol_6, Pol_7, Pol_8, Pol_9) are highest in their domain, further supporting discriminant validity. The HTMT ratio for Policy Support with other domains is below the threshold of 0.9, confirming that the constructs are distinct.

A comprehensive 40-item statement based on the SEM of treatment behavior for MHD patients was initially developed. During the EFA, four items were removed due to redundancy and low loadings, resulting in 36 items. Further refinement in the measurement model led to the removal of an additional five items with low loadings. Ultimately, 31 items were retained, forming four distinct domains that effectively capture the multifaceted nature of treatment behavior in MHD patients.

## Discussion

This study aimed to develop and validate an Arabic version of an SEM-based questionnaire to assess multi-level support factors influencing patients undergoing MHD. The findings demonstrate that the instrument possesses strong psychometric properties across content, face, structural, convergent, and discriminant validity. These results suggest that the tool is both theoretically sound and practically applicable in Arabic-speaking contexts.

The content validity indicators exceeded accepted thresholds, with I-CVI values ranging from 0.83 to 1.00, and scale-level indices (S-CVI/Ave and S-CVI/UA) between 0.98 and 1.00 and 0.90 and 1.00, respectively [34]. Face validity was also confirmed, with an average ICC of 0.963, indicating excellent agreement among patients [35]. Importantly, grammar and formal structure were reviewed by linguistic experts during content validation, while patient input during face validity focused on clarity and comprehensibility.

The four-factor structure identified via EFA aligned with the interpersonal, community, organizational, and policy domains of the SEM. Notably, the Community Support domain emerged as the most robust construct, consistent with the SEM's emphasis on mesosystem influences. In fragile health systems like Gaza's, community-based networks often step in to fill systemic gaps left by under-resourced institutions. This aligns with Bronfenbrenner's ecological theory, which conceptualizes community systems as key mediators between the individual and broader structural forces. Similar patterns have been observed in SEM-based studies in other conflict-affected or resource-constrained regions, such as Nigeria and South Asia, where local networks were found to provide more actionable support than formal structures [9,13].

The Interpersonal Support domain, primarily encompassing family and close social connections, also showed high reliability and convergent validity. This reaffirms findings from other chronic illness populations (e.g., diabetes, heart failure) where familial support strongly predicted adherence, emotional coping, and quality of life [22,36]. However, unlike some Western validation studies of SEM [11], where interpersonal support was more moderately associated with patient outcomes, its pronounced effect here reflects the cultural context of Arab collectivist societies, where family is central to healthcare navigation and decision-making.

The Organizational Support domain displayed moderate performance, with acceptable reliability ($\alpha = 0.857$) but borderline AVE (0.526), suggesting that while patients recognize healthcare institutions as important, their expectations or experiences may be inconsistent. This mirrors findings from a previous validation study [37], which also showed variability in organizational support perception for identifying delays in treatment for breast cancer patients. The implication is that organizational support may be a more dynamic and structurally sensitive construct, influenced by variations in staff training, communication, and service integration.

The Policy Support domain was the weakest performer, with AVE barely above the threshold and multiple items showing low factor loadings (e.g., Pol_7 = 0.562; Pol_1 = 0.576). Theoretical explanations for this include the "perceptual distance" between patients and macro-level influences in settings where public policy is either fragmented, politicized or largely inaccessible. In Gaza, this issue is particularly acute. The blockade, internal political divisions, and chronic humanitarian crises have undermined trust in public systems and led to disengagement from state-led initiatives. These contextual factors likely influenced respondents' perceptions of policy effectiveness—making policy support appear more abstract or irrelevant in their lived experiences. The findings are consistent with research showing that policy frameworks and government support significantly impact the care and outcomes of patients on chronic treatments such as MHD [38].

While the instrument demonstrated strong discriminant validity—confirmed through the Fornell-Larcker criterion, cross-loadings, and HTMT ratios—some minor conceptual overlap remains, particularly between community and organizational domains. This is not unexpected, as boundaries between informal and formal support can be blurred in contexts where civil society and public services intersect (e.g., NGOs operating inside hospitals). confirmatory factor analysis (CFA) was not conducted as the present study was inherently exploratory. Future research should apply CFA to independently test and confirm the factor structure identified here, thus strengthening the evidence for the instrument's validity.

A notable limitation of this study is the limited predictive ability of the Policy Support factor. One explanation is that in Arabic-speaking and Middle Eastern contexts, especially in regions such as Gaza, policy-related items may be perceived as abstract or less actionable due to patients' limited exposure to consistent health policy, fragmented health governance, or shifting policy landscapes caused by ongoing sociopolitical instability. This often results in insufficient knowledge, awareness, or engagement with policy initiatives among patients, which can reduce the perceived relevance or clarity of policy-related questions on the questionnaire. Moreover, in such settings, patients' daily experiences are shaped more by immediate interpersonal and community networks than by institutional or policy-level influences, creating a "perceptual distance" from macro-level policy actors. Low health literacy regarding formal health policies and generalized mistrust in governmental systems are additional factors known to attenuate the impact of policy support measures in patient-reported outcomes, as noted in prior international and regional research [39]. Additionally, the strong collectivist orientation in Arab societies places greater value on family and community support; the family is often the central source of practical, emotional, and informational assistance in the management of chronic disease, including hemodialysis. This cultural emphasis may further limit the salience of system-level policies in daily health management, as observed in studies across the Arab world and other Muslim-majority cultures. Consequently, "policy support" as a construct may not capture influences that are considered meaningful or effective within such cultural environments, contributing to the observed lower factor loadings and weaker explanatory power [40]. Future investigations should use qualitative and participatory research to further explore how Arab hemodialysis patients interpret and engage with health policy, refining the language and scope of policy-related items to better resonate with local beliefs and lived experiences.

While this study focuses on the interpersonal, organizational, community, and policy levels of the SEM, the intrapersonal domain was not explicitly measured through a dedicated questionnaire. Instead, socio-demographic variables (e.g., age, gender, education level, income, marital status) were considered as proxies for the intrapersonal domain [23]. This approach is supported by prior research, which suggests that socio-demographic characteristics often reflect individual-level factors such as beliefs, attitudes, and self-efficacy, which are central to the intrapersonal domain [41]. However, it is important to acknowledge that socio-demographic variables may not fully capture the complexity of

intrapersonal factors and could instead act as moderating variables that influence the relationship between other SEM levels and QoL outcomes [42].

Subsequent studies should undertake cross-validation of this instrument across diverse subgroups within the hemodialysis populations such as by gender, education level, or marital status—to assess the invariance and reliability of the questionnaire in different demographic segments. Additionally, there is a critical need to extend validation efforts to other Arab countries, as cultural, socioeconomic, and healthcare system differences may influence the relevance and interpretation of SEM constructs. Such multi-site, cross-national studies would provide a more robust understanding of the questionnaire's applicability throughout the broader Arabic-speaking world. Incorporating CFA in these future studies, alongside longitudinal assessments of reliability, will further reinforce the instrument's structural and predictive validity.

Although several limitations, this study provides the first Arabic-language SEM-based questionnaire validated for use among MHD patients. By integrating robust psychometric evaluation with theoretical grounding, it offers a novel tool for assessing social and structural influences on patient outcomes. Beyond research, the tool holds significant value for clinical screening, program evaluation, and health policy formulation in Arabic-speaking and conflict-affected populations.

## Conclusion

This study presents the development and validation of the Arabic version of the SEM questionnaire for patients undergoing MHD. The findings demonstrate strong content, face, and construct validity, with the four identified support domains—community, interpersonal, organizational, and policy—showing varying degrees of psychometric strength. Among them, Community Support emerged as the most robust construct, reflecting the critical role of social networks in the well-being of MHD patients. Interpersonal and Organizational Support also demonstrated high reliability, although a few items with low factor loadings were removed or flagged for future refinement. The Policy Support domain, while conceptually important, exhibited weaker validity, suggesting the need for improved measurement clarity and item alignment.

The final validated tool consists of 31 items and can serve as a culturally sensitive and psychometrically sound instrument for assessing the multi-level influences affecting MHD patients in Arabic-speaking settings (S1 File, S2 File). The questionnaire's strong discriminant validity reinforces its structural integrity and potential application in both research and clinical contexts. To further enhance its utility and generalizability, future studies are encouraged to retest the low-loading items, expand the sample size, and employ longitudinal designs to evaluate stability over time. By doing so, this instrument may contribute to the development of more targeted and effective interventions aimed at improving the quality of life and treatment adherence among MHD patients in the region.

## Supporting information

**S1 File. Developed English version of social-ecological model questionnaire for hemodialysis patients.**
(DOCX)

**S2 File. The arabic version of the socio-ecological model questionnaire for hemodialysis patients.**
(DOCX)

## Acknowledgments

We would like to convey our gratitude to the study participants and the Palestinian Ministry of Health for permitting us to conduct this study. We also extend our sincere appreciation to UNIMAS for their valuable support throughout the course of this study.

## Author contributions

**Conceptualization:** Ayman S. Abutair, Md Rahman, Asri Bin Said.

**Data curation:** Ayman S. Abutair.

**Methodology:** Ayman S. Abutair.

**Supervision:** Md Rahman, Asri Bin Said.

**Validation:** Ayman S. Abutair.

**Writing – original draft:** Ayman S. Abutair.

**Writing – review & editing:** Ayman S. Abutair, Md Rahman, Asri Bin Said.

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
