## [Decision Letter · Decision Letter 0]

22 Apr 2025

Dear Dr. Abutair,

Thank you for submitting your manuscript to PLOS ONE. After careful consideration, we feel that it has merit but does not fully meet PLOS ONE’s publication criteria as it currently stands. Therefore, we invite you to submit a revised version of the manuscript that addresses the points raised during the review process.

We look forward to receiving your revised manuscript.

Kind regards,

I Gede Juanamasta

Academic Editor

PLOS ONE

3. Please include a copy of Table 1-4 which you refer to in your text on page 8-9.

Additional Editor Comments (if provided):

Reviewers' comments:

Reviewer's Responses to Questions

**Comments to the Author**

1. Is the manuscript technically sound, and do the data support the conclusions?

Reviewer #1: Yes

Reviewer #2: Partly

Reviewer #3: Yes

2. Has the statistical analysis been performed appropriately and rigorously?

Reviewer #1: No

Reviewer #2: No

Reviewer #3: Yes

3. Have the authors made all data underlying the findings in their manuscript fully available?

Reviewer #1: Yes

Reviewer #2: Yes

Reviewer #3: Yes

4. Is the manuscript presented in an intelligible fashion and written in standard English?

Reviewer #1: Yes

Reviewer #2: No

Reviewer #3: Yes

Reviewer #1: Introduction

The background information in the introduction on End-Stage Renal Disease (ESRD), the problems of Maintenance Hemodialysis (MHD) patients, and the relevance of the Social-Ecological Model (SEM) framework justify the need for developing a culturally sensitive instrument to assess multi-level social influences among MHD patients in Arabic-speaking communities.

• Gap identification is lengthy: The introduction takes too long to state general SEM applications before it gets to the specific gap this research aims to address. Move the study's aim and justification forward to improve the logical flow.

• Limited application to MHD-specific issues: While SEM is presented, its utility in the context of MHD patients (access to healthcare, medication adherence, psychosocial impact) has not been fully developed. Add more MHD-specific sources (e.g., hemodialysis social support and quality of life research) to support the rationale.

• Absence of regional framework: The importance of investigating Arabic-speaking MHD populations, owing to cultural and systemic healthcare variability, might be more clearly articulated.

Literature Review

Literature review is included in the introduction and is centered primarily on the general use of SEM in health promotion. There is little detail about existing tools or studies measuring social-ecological factors in MHD or chronic disease environments.

• Limited scope: There is a lack of coverage regarding SEM usage in chronic disease or MHD contexts. Add references on SEM usage in managing chronic disease and studies on dialysis that examine social support systems.

• No reference to other measurement instruments: The article would be more robust with a consideration of other scales measuring social or ecological factors within health settings. Reference scales like the MOS Social Support Survey or other SEM tools that have been validated to place the new instrument in literature.

• Cultural gap: The review does not mention social and healthcare dynamics in Gaza or other similar regions that justify the tool's development. Cite regional studies to make it more relevant.

Methodology

The methodology section describes the design of the questionnaire, translation, testing of content and face validity, data gathering from Gaza MHD patients, and psychometric testing. ICC for face validity, Exploratory Factor Analysis (EFA) with Varimax rotation, convergent validity (AVE, CR), and discriminant validity (Fornell-Larcker, cross-loadings, HTMT) are given in the article.

• Sample size not provided for EFA: Nowhere it is mentioned which particular sample size has been used in performing the EFA. Specify sample size and explain on the basis of psychometric grounds (min. 5-10 respondents per item).

• Inadequate description of sampling procedure: "Systematic random sampling" is not clear. Specify the sampling interval and how systematic sampling was carried out in dialysis centers.

• No explanation of missing data treatment: Nothing is mentioned about how incomplete or missing answers were treated. Add a data cleaning and missing data treatment section.

• Exclusion of intrapersonal level buried: The explanation for the exclusion of intrapersonal factors is buried beneath preliminary results. Put this design decision in the conceptual framework subsection of Methods and support it with literature.

• Assumptions of EFA not tested or reported. No KMO; Bartlett's test; Factorability indices. Include KMO and Bartlett's test to establish if EFA is appropriate.

• Multicollinearity problem: A possible issue not addressed. Conduct VIF analysis or use Principal Axis Factoring (PAF) instead of Varimax/Principal Components.

• Rotation choice not justified: Varimax assumes orthogonal factors, possibly not consistent with SEM theoretical frameworks. Use oblique rotation (Promax) if constructs ought to correlate.

• Low-loading items treated opaque: Removal item criteria not defined. Specify threshold used for item deletion (e.g., <0.4 loading) and provide a pre/post-removal report.

• Normality of data for ICC not tested: ICC calls for normally distributed ratings. Provide normality tests (e.g., Shapiro-Wilk) and discuss ICC model selection.

Discussion

The analysis restates the key findings and backs them with general references but is thin on interpretation and theoretical analysis.

• Over-reliance on replication of findings without integrating broader literature. Enhance the discussion by contrasting the findings with a prevailing SEM validation study or social support study in chronic illness.

• Policy Support issues underanalyzed: The likely causes of poor performance are not considered. Consider if cultural or political circumstances in Gaza influenced this aspect.

• Cross-sectional design flaws barely touched on: No word on test-retest reliability. Recommend longitudinal studies or stability tests in the future.

• Replace the bullet-point "Takeaway Messages" presentation and weave out key points into an integrated academic conclusion.

• Highlight useful applications of the tool by policy makers or practitioners.

Reviewer #2: Thank you for the opportunity to review this research. This study focuses on developing a tool based on the Social-Ecological Model for patients undergoing hemodialysis. I would like to draw the authors’ attention to the following comments:

Abstract

• In the methods section of the abstract, first mention the study design.

• This sentence in the methods section of the abstract needs revision because item analysis, content validity assessment, and face validity evaluation are essentially considered psychometric indices. Therefore, after mentioning these three, you should specify which other psychometric indices were assessed:

"involved item analysis, content validity assessment, face validity evaluation, and psychometric analysis."

• The methods section of the abstract needs better organization: When you mention which psychometric indices were evaluated, you should also specify the method used for their assessment at the same point. Avoid scattered explanations.

• This sentence is unclear: For which variable was the intraclass correlation coefficient calculated?

"Face validity was evaluated using Intraclass Correlation Coefficient analysis."

• The methods section of the abstract should mention the study population, sample size, and sampling method.

• The findings in the abstract should be reported in the same order as the indices were examined in the study. This consistency should also be maintained in the methods section.

• The following sentences belong to the findings section, not the methods. In the methods section, only mention the analysis method:

"Exploratory Factor Analysis identified four distinct support dimensions: family and interpersonal support, community support, organizational support, and policy support, explaining 65.15% of the total variance."

• The methods section of the abstract does not mention the assessment of convergent and discriminant validity, yet they are reported in the findings. This inconsistency should be corrected.

• The following sentences are confusing. Are you referring to subscales? It would be better to first present the overall factor analysis results and then report the subscale findings:

"Community Support exhibited the strongest convergent validity, with high reliability and explained variance. At the same time, family and interpersonal support and organizational support were reliable but contained low-loading items that required attention."

• The conclusion stated in the abstract does not align with the study’s objective. The conclusion should be based on the study’s aim and results:

"The study highlights the importance of social and institutional support in MHD patient care."

• It would be better to include keywords related to validity, reliability, and psychometrics.

Introduction

• This phrase needs better wording:

"heightened psychological distress"

• Who introduced the Social-Ecological Model? What dimensions does it include? Provide an explanation.

• The introduction lacks a discussion on the impact of applying the Social-Ecological Model. What role can this model play in patients undergoing hemodialysis?

• Which studies have examined the effectiveness of this model, leading you to develop a specific assessment tool for it?

• Are there existing tools for this purpose? What tools are currently used to evaluate the effectiveness of this intervention? What are their limitations?

• What cultural factors in Arabic-speaking populations necessitate a localized assessment tool? These aspects should be addressed in the problem statement.

Methods

• At the beginning of the methods section, introduce a subsection titled "Design" and describe the study design, research setting, and time frame.

• Define a subsection titled "Participants and Sampling Method" and describe the study population, inclusion and exclusion criteria, sampling method, and sample size calculation.

• The questionnaire development process is insufficiently explained. Did you use existing tools to generate the item pool? How were these tools searched? The full search process should be described, including keywords, databases, search timeframe, search language, and the types of studies included. A PRISMA flowchart should be provided.

• Why was a qualitative study not used to generate items?

• Face validity should be assessed before content validity.

• Why was the item impact score not calculated for face validity?

• Cite a reference for the face validity assessment method used.

• Specify the expertise of the individuals who evaluated content validity.

• Why was the kappa coefficient not calculated after assessing content validity?

• Can patients assess grammar and language? This should be evaluated in content validity and verified with experts. Additionally, item redundancy should be examined.

• The heading "Psychometric analysis" should be moved before the content validity assessment and replaced with "Structural Validity," where factor analysis, convergent validity, and discriminant validity should be discussed.

• What assumptions were checked before conducting factor analysis?

• Why was confirmatory factor analysis (CFA) not performed?

• Why was the reliability and internal consistency of the tool not assessed?

• The explanations under "Psychometric Analysis" should be moved to the study design section at the beginning of the methods section.

• Was the study’s objective:

"to investigate the multifaceted influences of sociodemographic characteristics, interpersonal and community factors, organizational issues, and policy matters on nutritional status, psychological distress, and the QoL of patients undergoing maintenance HD in the Gaza Strip"

or to develop a tool for measuring social-ecological dimensions? This should be clarified, as the title and abstract mention tool development.

• Provide a detailed explanation of the systematic random sampling method used.

• This sentence is unclear:

"The collected data were subjected to rigorous manual examination and validation protocols to ensure precision."

• The statistical analysis section contains redundant information that was already mentioned in the methods section. Remove duplicate content.

Results

• In demographic results, report both percentages and absolute numbers.

• Up to this point in the manuscript, the number of patients included in the study has not been specified!

• Item analysis was not mentioned in the methods section, yet it appears as a heading in the results section.

• How many items were initially generated for the item pool? How many items remained in the final psychometric evaluation tool?

• The number of items in each stage of analysis and the number retained after each stage must be reported in the results section.

• Describe how factor analysis was conducted, what criteria were used to determine factors, what criteria were used to retain items, and how sample adequacy was confirmed.

• Why was PCA used instead of EFA?

• Why was Varimax rotation, an orthogonal rotation method, applied?

• What was the sample size for factor analysis?

• In Table 3, include communalities for each item along with factor loadings, and report eigenvalues and explained variance for each factor.

Reviewer #3: April 14, 2025

Reviewer Report

Dear Editor,

Thank you for the opportunity to review this manuscript titled “Development and Validation of the Arabic Version of the Social-Ecological Model Questionnaire for Patients Undergoing Maintenance Hemodialysis.” I have carefully read through the manuscript and provide my review below.

In summary, this is a validation study that aimed to assess the psychometric properties of an Arabic questionnaire assessing structural and social determinants of health within the framework of the Social-Ecological Model (SEM). The authors report on the process of questionnaire development, validation, including translation and administration of the questionnaire to a sample of patients recruited from all governmental hemodialysis centers in the Gaza Strip, Palestine. The scale included 10 items for each SEM domain: interpersonal, organizational, community, and policy factors. The authors extracted four factors using Exploratory Factor Analysis, which accounted for a significant amount of variance in the sample. They also report on the validity of their constructs.

This work is important as it provides clinicians and policymakers in the Arab world with a tool to assess the social and structural factors that undoubtedly influence illness and impact the management of individuals on maintenance hemodialysis. I have also reviewed the Arabic questionnaire, which I believe is a valuable addition to the tools available for supporting patients undergoing maintenance hemodialysis.

I believe the current manuscript would benefit from additional details regarding the sampling technique, recruitment sites, and the number of included patients, both in the main text and the abstract. I also recommend that the authors provide references for the sources used in developing the questionnaire items, which includes any other questionnaires or scales measuring the same constructs.

Thank you again for the opportunity to review this manuscript.

**Do you want your identity to be public for this peer review?** For information about this choice, including consent withdrawal, please see our Privacy Policy

Reviewer #1: **Yes: ** Stefanos Balaskas

Reviewer #2: No

Reviewer #3: No

---

## [Author Response · Author response to Decision Letter 1]

30 May 2025

We thank the Academic Editor and all reviewers for their valuable comments, which helped us improve the clarity, rigor, and scientific merit of our manuscript. We provide a detailed, point-by-point response to each comment. Changes made in the manuscript are highlighted in the revised version with track changes, and corresponding line numbers are provided where applicable.

---

## [Decision Letter · Decision Letter 1]

2 Jul 2025

Dear Dr. Abutair,

Thank you for submitting your manuscript to PLOS ONE. After careful consideration, we feel that it has merit but does not fully meet PLOS ONE’s publication criteria as it currently stands. Therefore, we invite you to submit a revised version of the manuscript that addresses the points raised during the review process.

We look forward to receiving your revised manuscript.

Kind regards,

I Gede Juanamasta

Academic Editor

PLOS ONE

Reviewers' comments:

Reviewer's Responses to Questions

**Comments to the Author**

Reviewer #1: (No Response)

Reviewer #3: All comments have been addressed

2. Is the manuscript technically sound, and do the data support the conclusions?

Reviewer #1: Partly

Reviewer #3: Yes

3. Has the statistical analysis been performed appropriately and rigorously?

Reviewer #1: No

Reviewer #3: Yes

4. Have the authors made all data underlying the findings in their manuscript fully available?

Reviewer #1: Yes

Reviewer #3: Yes

5. Is the manuscript presented in an intelligible fashion and written in standard English?

Reviewer #1: No

Reviewer #3: Yes

Reviewer #1: Introduction

The background offers a general justification for the research, highlighting the need for psychosocial support during hemodialysis and the necessity of context-specific measurement tools among Arabic-speaking populations. It uses the Social-Ecological Model (SEM) as the theory to inform the development of the questionnaire.

• The introduction can also be enhanced by the clear stipulation of research questions and goals, ideally as a last paragraph to ground the study.

• The reasoning for the application of SEM in particular—apart from its multi-level aspect—must be justified with excerpts of previous empirical use of SEM in treatment of chronic disease or dialysis. Include a concluding paragraph in the introduction that outlines study aim, objective, and research importance in a clear manner to maintain greater coherence and reader direction.

Literature Review

The literature review tells us about the SEM and the hemodialysis burden but fails to link these observations to each other or to place properly the need for a new instrument or adaptation.

• The relationship between SEM areas and burden of dialysis is under-explored. For example, what effect do organizational-level support have on outcomes for MHD? Incorporate empirical studies that utilize SEM to investigate compliance or chronic illness management (e.g., HIV, diabetes, ESRD).

• Lacking regional specificity—no distinct references to existing health literacy or psychosocial research on Arabic-speaking hemodialysis patients. Add Arab-world or Middle Eastern-specific research of support systems or barriers in treatment.

• Neglects to mention comparable validated instruments (e.g., MOS-SSS, MSPSS) and how the SEM-based instrument varies or is superior. Provide a concise critique of current measures of assessment and reasoning of support, which account for why a SEM-based measure is theoretically superior or more fitting.

Methodology

The research employs the typical psychometric validation process consisting of content validity, face validity, exploratory factor analysis (EFA), reliability, and convergent/discriminant validity testing.

• The cultural translation and adaptation are not provided with critical details. For example, how variations in forward–backward translation were managed is not specified. Define translator qualifications and whether pilot test or panel review was done.

• ICC face validity testing is not adequately described; there is no ICC model (e.g., ICC(2,1)) and no confidence intervals. Identify the ICC model used, include confidence intervals, and explain threshold interpretations.

• Sampling strategy is not justified—no reason why the sample size was chosen or whether power was taken into account. Clarify the sample size on the basis of acceptable psychometric practice (e.g., participants per item, power analysis).

Statistical analyses include descriptive statistics, EFA, and reliability/validity statistics. These are suitable for stage-one scale validation but are short of some assumption testing and transparency.

• EFA's assumptions (e.g., multivariate normality, linearity) are not tested or stated. Test and report multivariate normality if Maximum Likelihood extraction is employed.

• Use parallel analysis or scree plot inspection to make an argument for number of factors retained.

• Report correlation and factor loading matrices to allow transparency.

• Retention of factors is eigenvalue and explained variance based only. Parallel analysis and other stricter criteria are not employed.

• Fornell–Larcker and HTMT thresholds are noted but not interpreted.

• Place all validity/reliability measures into a single table for ease of interpretation.

• Tabular summaries of factor loadings and communalities are not presented.

• Graphical aids (e.g., scree plot, histograms of item distributions) are not included to aid interpretation.

• Measurement Invariance: No subgroup analysis (e.g., gender, age) even with an imbalanced sample.

• Minimal explanation of why certain items (e.g., Pol_10, Org_5) underperformed and conceptual implications.

Discussion and Conclusion

The discussion is in terms of the study purposes and SEM model and recognizes novelty and usability of the validated tool but is devoid of theory and limitation consideration.

• The limited predictive ability of the "Policy Support" factor is not addressed in detail. Explain why questions related to policy may have underperformed (e.g., cultural confusion, insufficient policy knowledge).

• Low degree of engagement with cultural or systems-level implications of results. Consider cultural conceptions of support of Arab hemodialysis patients.

• Limitations section can be amended to detail the sample's features (e.g., gender bias) and cross-validation requirements. Suggest a cross-group validation (e.g., education, gender) and confirmatory factor analysis (CFA) in subsequent research.

• Confirmatory Analysis: Because it is exploratory research, CFA is understandably missing. Nevertheless, this should be discussed and suggested to be performed in the future.

Reviewer #3: Thank you for the opportunity to review this manuscript. I believe all comments have been addressed.

**Do you want your identity to be public for this peer review?** For information about this choice, including consent withdrawal, please see our Privacy Policy

Reviewer #1: **Yes: ** Stefanos Balaskas

Reviewer #3: No

---

## [Author Response · Author response to Decision Letter 2]

1 Sep 2025

We have addressed all the comments and suggestions provided by the reviewers. Any discrepancies have been clearly explained in the rebuttal letter. Please let me know if there is anything further we can do to improve the article.

---

## [Decision Letter · Decision Letter 2]

17 Sep 2025

Development and Validation of the Arabic Version of the Social-Ecological Model Questionnaire for Patients Undergoing Maintenance Hemodialysis

PONE-D-25-10867R2

Dear Dr. Rahman,

We’re pleased to inform you that your manuscript has been judged scientifically suitable for publication and will be formally accepted for publication once it meets all outstanding technical requirements.

Kind regards,

I Gede Juanamasta

Academic Editor

PLOS ONE

Additional Editor Comments (optional):

Reviewer #1:

Reviewers' comments:

Reviewer's Responses to Questions

**Comments to the Author**

Reviewer #1: All comments have been addressed

2. Is the manuscript technically sound, and do the data support the conclusions?

Reviewer #1: Yes

3. Has the statistical analysis been performed appropriately and rigorously?

Reviewer #1: Yes

4. Have the authors made all data underlying the findings in their manuscript fully available?

Reviewer #1: Yes

5. Is the manuscript presented in an intelligible fashion and written in standard English?

Reviewer #1: Yes

Reviewer #1: The authors have addressed all my comments and concerns, I believe the paper is ready for publication.

**Do you want your identity to be public for this peer review?** For information about this choice, including consent withdrawal, please see our Privacy Policy

Reviewer #1: **Yes: ** Stefanos Balaskas

---

## [Editor Report · Acceptance letter]

PONE-D-25-10867R2

PLOS ONE

Dear Dr. Rahman,

I'm pleased to inform you that your manuscript has been deemed suitable for publication in PLOS ONE. Congratulations! Your manuscript is now being handed over to our production team.

Kind regards,

on behalf of

Dr. I Gede Juanamasta

Academic Editor

PLOS ONE